# Porous Graphene Oxide Decorated Ion Selective Electrode for Observing Across-Cytomembrane Ion Transport

**DOI:** 10.3390/s20123500

**Published:** 2020-06-21

**Authors:** Shihui Hu, Rong Zhang, Yunfang Jia

**Affiliations:** College of Electronic Information and Optical Engineering, Nankai University, Tianjin 300071, China; 1120170101@mail.nankai.edu.cn (S.H.); 1120190126@mail.nankai.edu.cn (R.Z.)

**Keywords:** porous graphene oxide, interfacial micro-environment, ion selective electrode, ion transport, sodium iodide symporter

## Abstract

The technology for measuring cytomembrane ion transport is one of the necessities in modern biomedical research due to its significance in the cellular physiology, the requirements for the non-invasive and easy-to-operate devices have driven lots of efforts to explore the potential electrochemical sensors. Herein, we would like to evidence the exploitation of the porous graphene oxide (PGO) decorated ion selective electrode (ISE) as a detector to capture the signal of cytomembrane ion transport. The tumor cells (MDAMB231, A549 and HeLa) treated by iodide uptake operation, with and without the sodium-iodide-symporter (NIS) expression, are used as proofs of concept. It is found that under the same optimized experimental conditions, the changed output voltages of ISEs before and after the cells’ immobilization are in close relation with the NIS related ion’s across-membrane transportation, including I^−^, Na^+^ and Cl^−^. The explanation for the measured results is proposed by clarifying the function of the PGO scaffold interfacial micro-environment (IME), that is, in this spongy-like micro-space, the NIS related minor ionic fluctuations can be accumulated and amplified for ISE to probe. In conclusion, we believe the integration of the microporous graphene derivatives-based IME and ISE may pave a new way for observing the cytomembrane ionic activities.

## 1. Introduction

The across-cytomembrane ion transport is one of the important issues in the cellular metabolism. There are anomalous episodes in the growth and apoptosis of tumor cells [1], for example, the uncoupled K^+^ and Cl^−^ flows in the process of regulatory volume decrease, which were found in nasopharyngeal carcinoma CNE-2Z cells [2], and the differences of Na^+^ and K^+^ transportation in the weakly and highly metastatic prostatic epithelial carcinoma cell lines [3]. Moreover, it is also found in the studies of cancer therapy, that the increased iodide uptake induced by the transfected sodium iodide symporter (NIS) on tumor cells was helpful to a targeted radiotherapy [4], and that the artificial cytomembrane influx of Cl^−^ can be trigged by the released squaramide at the tumor site and increase the effect of homeostatic perturbation therapy [5]. With the deep understanding of cytomembrane ion channels which are involved in almost all of the oncological stages and treatments [6], the requirement to observe their influences on ions’ distributions promotes the emergence of state-of-the-art technologies in this area. 

In this aspect, the patch clamp technique is a conventional method, which can convert the ion flux to measurable currents [7]. It has provided substantial evidence in the identification of the accesses for ions [8], such as potassium [9], chloride [10], proton [11], etc., as well as in the interpretation of the light-controllable ions’ transportations gated by Channelrhodopsins (ChR) [12]. Recently, its integration with fluorescence labelling method [13] has proven to be able to simultaneously detect the transmembrane current and fluorescence intensity [14]. Meanwhile, the combination with microfluidic approaches made great possibilities for building the wafer-based patch-clamp device [15], which was utilized to record the unique channel signatures of endogenous chloride and potassium channels in HEK 293T cells [16]. 

From the perspective of the non-invasive and selective detection, the electrochemical sensors are also good candidates [17], both the ion-selective field effect transistors (ISFETs) [18,19] and ion selective electrodes (ISEs) [20,21] have been developed to observe the transmembrane ions’ movement. In these researches, an interfacial micro-environment (IME) between cells and sensors was fabricated by the micromachining techniques. Not only the weak ionic variations in IME, which were caused by the ion transport across the epithelial cell layer, like the fluxes of Cl^−^ [18] and K^+^ [19], could be detected, but also the multi-ions’ detection of Na^+^, K^+^, Cl^−^ and pH could be realized [20,21]. However, for I^−^, whose uptake and outflow mediated by NIS are of great significance for cancer treatment [4], it is still the minority in the detection of transmembrane ion transport. As far as we know, the current observation methods for I^−^ transport are mainly focused on radio-assay [22] and fluorescent labelling method [23], no electrochemical contribution has been found.

In our opinion, no matter what kind of electronic devices were used in previous research, the means to form the IME, like the micropipette in the patch clamp technique [7], or the micro-fabricated space on ISFET [19] or ISE [20], were prerequisite. Because the transmembrane I^−^ flow is weak, the measuring sensor and the concerned cell should be kept at a distance as close as possible, so as to focus the fluctuation of I^−^ concentration (C_I_) on the sensing surface, and not affect the sensor’s original sensitivity. Herein, we would like to propose an attempt in this area by using ISE because of its merits in non-invasive and non-cytotoxicity in contrast to the patch clamp technique [17].

As illustrated in Figure 1A, a porous graphene oxide (PGO) scaffold IME is built on the iodide sensitive ISE (I^−^-ISE) with the sensing film of Ag_2_S/AgI. In our previous work, it was proven that the PGO decoration on ISE could provide a robust three-dimensional (3D) and biocompatible media for the immobilized cells [24]. We conjecture (as depicted in Figure 1B) that the PGO decorated I^−^-ISE (PGO-ISE) could be developed as an alternative device for observing the ionic fluctuation induced by the across-cytomembrane ion transport, i.e., the different I^−^ distributions between the cells in the tested group (TG) and the control group (CG), which are with and without NIS expression on their cytomembrane, respectively. The tumor cells of MDAMB231, A549, HeLa are used as proof-of-concepts [4], the over-expressed nucleolin surface sites on their membranes [25] can be utilized to fix them on the aptamer (nominated as AS1411) functionalized PGO-ISE (AS1411-PGO-ISE), based on the immuno-affinity. Meanwhile, these cells in each group are different from being treated by the iodide-uptake operation (IU), or not (NIU), to invoke the variation of the cytosolic I^−^ ions. Ultimately, the AS1411-PGO-ISEs’ output voltages (V_out_) are recorded before and after the cells’ immobilization, under similar background solutions (i.e., the KI/KNO_3_ buffer solutions with different concentrations).

## 2. Materials and Methods

### 2.1. Materials

The cells’ suspensions were all purchased from Tianjin Saier Biotechnology Co., Ltd. (China), including MDAMB231, A549 and HeLa cells with and without NIS, and treated by the iodide uptake for 0, 30, 60, 90 and 120 min according to our design, named as NIU, IU30, IU60, IU90 and IU120, respectively. The sequence of AS1411 (Sangon Biotech Co. Ltd., China) is: 5’-NH_2_-(CH_2_)_6_-GGT-GGT-GGT-GGT-TGT-GGT-GGT-GG-3’. The AS1411 was diluted by the buffer (10 mM Tris, 2.5 mM MgCl_2_, 140 mM KCl, pH 7.4), then the obtained stock solution was adjusted to 1 μM for the following experiments. The other reagents used in the experiments were: (1) Piranha solution, prepared by 30 mL H_2_SO_4_ (98.0%) and 10 mL H_2_O_2_ (30.0%). (2) 3-aminopropyltriethoxysilane (APTES) in DIW with the volume ratio (*V*/*V*) of 1:10, pH = 7.4. (3) glutaraldehyde (GA) in DIW (*V*/*V* = 1:20), Ph = 7.4. (4) The phosphate buffer saline (PBS) is Na_2_HPO_4_ and NaH_2_PO_4_ in DIW,100 mM and pH 7.4.

### 2.2. Principle of I^−^-Ion Selective Electrode

The schematic diagram of ISE is depicted in Figure 1A, the detailed description about the principle of ISE can be found in our previous work [24]. According to the Nernst equation, ISE’s V_out_ is in a linear relationship with the logarithm of the I^−^ ions’ concentration (C_I_) at the interface between the Ag_2_S/AgI crystal film and the solution (as diagramed in Figure 1A). V_out_ is the voltage caused by the concentration difference across Ag_2_S/AgI, therefore V_out_ is in a positive dependence on C_I_. In the practical application, the standard curve method is used to obtain C_I_ by the measured V_out_.

### 2.3. Synthesis of Porous Graphene Oxide

The preparation of PGO was conducted by the following process. (1) The graphene oxide (GO) was prepared from graphite powder by the modified Hummers method. (2) The suspended GO/DIW solution was reacted with thiourea in the autoclave (180 °C, 4.5 h), to mediate the GO flakes into small lamellae, then they were crosslinked with each other to form the porous structure. (3) The PGO suspension (0.05 mg/mL) was prepared by dispersing in DIW, and treated by ultra-sonication for 1–2 h.

### 2.4. ISE’s Modification

The ISE’s modification processes were: (1) After washing with DIW, the naked ISE was immersed in APTES solution (50 °C, 2 h) and GA solution (room temperature, 1 h) in sequence; (2) 150 μL 0.05 g/mL PGO solution was dripped on the APTES/GA treated ISE’s surface, and dried overnight at room temperature; (3) 100 μL EDC/NHS (200 mM/50 mM) was dripped on the PGO decorated ISE’s surface for 30 minutes, then rinsed by DIW; (4) 50 μL AS1411 (1 μM) was incubated on the modified ISE at 4 °C for 12 h; (5) 50 μL cell suspension was added on the surface of AS1411-PGO-ISE, after being incubated for 30 min at 37 °C and washed by DIW, the cell fixed ISE was ready for use.

### 2.5. Electrical Measurements

The ISEs in different experimental stages, including the bare, PGO decorated and AS1411 functionalized ISEs, as well as the different cells fixed ISEs were measured by a similar operation, that is immersing the bottom of the electrode into similar KI/KNO_3_ solutions, and connecting it with the measuring instrument.

### 2.6. Instruments

S-3500N (Hitachi, Japan) and WYS-CX23 (KJ Biotech, Tianjin, China) were used for the scanning electron microscope (SEM) and microscopy, respectively. LK3100D (Tianjin Lanbiao Electronic Technology Development Co., Ltd. China) was used for the electrical measurement. 

## 3. Results

### 3.1. Microscopy of PGO and Immobilized Cells on It

The micro-structure of the PGO layer was characterized by SEM, as shown in Figure 1C. The porous architecture could be identified with the size of several to tens of micrometers. After that, the microscopy was conducted in order to confirm that the cells can be immobilized on the AS1411 and PGO modified slides, regardless of whether they are the TG cells or the CG ones, or of whether they suffered the IU treatment or not. For these purposes, the microscopic photos were arrayed in two groups of TG and CG, the images in each group were also labelled according to the durations of IU, as shown in Figure 2.

Firstly, to recognize the fixed cells, the photos of the slides without cells (named as Blank in Figure 2) were used as references, and the identified cells in the other images were marked by red circles. Secondly, a comparison of the photos in the two groups indicated that there was no obvious difference between them. Thirdly, the possible influence caused by IU on cells’ immobilization was also examined by the varied lengths of this process (0, 30, 60, 90 and 120 min), as illustrated by NIU and IU30~IU120. There was no observable deviation in the quantity of cells between them, either. Accordingly, the microscopy examination indicated the quantity of the fixed cells was not affected by the expression of NIS and the IU operation, so its potential impact on changing V_out_ can be neglected.

### 3.2. I^−^ Responses of PGO-ISEs and Cells’ Immobilized Ones

The electronic responses of PGO-ISEs for I^−^ were examined before and after cells’ immobilizations, in which the cells (MDAMB231, A549 and HeLa) are the TG and CG ones, as well as different in with and without IU pre-treatments. The measured V_out_ data for TG (orange symbols) and CG (blue symbols) at each of the C_B_ are plotted in the coordinates of V_out_ vs. -log_10_C_B_, in Figure 3. By comparing the fitted lines of PGO-ISEs before (dashed) and after (solid) cells’ immobilization, it could be found, though the I^−^ sensitivity is still possessed by the cells anchored electrodes, that the slopes (K) of the solid lines are lower than those of the dashed lines. This phenomenon agrees with the experimental result in our previous work [24]. What’s more, in this work, two NIS related issues are pointed out by comparing the changed data (ΔV_out_) and fitted lines of TG (orange) and CG (blue) fixed PGO-ISEs. The first is that bigger changes of the fitted lines’ slope (ΔK) could be found by the TG immobilized devices compared with the CG fixed ones. The second is the deviations of V_out_ (ΔV_out_, expressed by the right *y*-axis) caused by the cells’ immobilization were also different between the two groups, that is: The ΔV_out_ values of the TG cells (the red columns) are higher than those of the CG cells (the blue ones). Meanwhile, the higher columns are always observed at the lower C_B_ (i.e., the right of *x*-axis) for all the cells, though their exact values are different. These observed deviations are attributed to the variations of I^−^ concentration in IME which are caused by the different cells (whether they are TG or CG, as well as IU or NIU), more discussions are presented in the following section.

### 3.3. Identification of PGO’s Function on ISE

The function of PGO in realizing the expected observations could be manifested by the similar tests conducted by the ISEs without the PGO decoration, as presented in Figure 4, in which the ISEs are only functionalized by AS1411 (named as AS1411-ISE). In comparison with Figure 3, it is found that the fitted responding lines of AS1411-ISEs are not obviously changed, that is to say, the solid and dashed lines of the same color in Figure 4A,B are not separated as in Figure 3. At the same time, the cells’ immobilization induced variations (ΔV_out_) of AS1411-ISEs (Figure 4C) are also distinct from the histogram data of the PGO decorated ISEs (Figure 3). In these situations, due to the absence of the PGO layer, the sensing surfaces of the ISEs are directly covered by the AS1411 anchored cells, and there is almost no space between them, the across-cytomembrane ion transport mediated C_I_ (as depicted in Figure 1B) cannot be established without the “spongy-like” buffer layer on the ISE (as mentioned in the introduction), therefore the AS1411-ISEs’ I^−^ responding data (V_out_) before and after the cells’ immobilizations are not obviously separated. It is deduced that the presence of the PGO layer is helpful to avoid the high density of the cells at the sensor surface, which can lead not only to the suppression of transmembrane transport but also to some kind of reuptake of I^−^ ions between the nearest cells.

However, there are still negative variations (the data columns of ΔV_out_ in Figure 4C), and we think it may be related to the negative influence on the sensing surface caused by the directly attached cells. At the same time, the positive columns at the lower C_B_ also indicate that there are varied interfacial I^−^ concentrations, but their values (less than 10 mV) are considerably lower than the data of similar PGO decorated ISEs in Figure 3A,D, which are about 55–70 mV.

### 3.4. Variations of I^−^ Concentration in IME

The understanding of the changed responding curves is proposed according to the different IME statues, as depicted in Figure 1B. Firstly, the increasement of V_out_ (in Figure 3 and Appendix A) suggests that, accompanied with the cells’ immobilization, there may be modulations to the interfacial I^−^ concentration (C_I_). Thereby, by using the standard curve method, we estimate C_I_ (C_TG_ and C_CG_) for MDAMB231, A549 and HeLa cells, shown in Figure 5A–C, in which, the negative indexes of the applied background I^−^ concentration(C_B_) and C_I_ are taken as the abscissas, the dashed line (y = x) indicates an imagined relation for C_I_ = C_B_. It could be found that the deviations between the data points and the dashed line are bigger on the right side.

We deduce the main reason for this phenomenon may be related to the ions’ activities induced by different cells, since the only changed element in each of Figure 5A–C is whether the cells are with or without NIS (TG or CG). As shown in the schematic illustration in Figure 1B, there are two kinds of ions’ behaviors, which are the ion diffusion under the concentration gradient from the background solution to the IME, and the across-cytomembrane ions transport which may be the movements of I^−^ caused by NIS or Cl^−^ due to its intra/extra-cellular difference. It can be found in Figure 5A–C that the offset of the data points deviating from the dashed line is at about 5 × 10^−4^ mol/L, which infers that when CB is higher than 5 × 10^−4^ mol/L, the ions in the IME are mainly controlled by the diffused I^−^ from the background solution, therefore CI is close to C_B_; on the contrary, under the lower C_B_ background, the ions’ transmembrane activities in the IME may be dominant, so that the deduced C_I_ are higher than C_B_, in other words, the measured V_out_ data increased, as shown in Figure 3. This means, for the purpose of observing the cells’ transmembrane ion transport, it would be better to make C_B_ at a lower degree. Meanwhile, according to Figure 3 and Figure 4, all the electrodes exhibited the same linear response curves, even at lower C_B_, no deteriorating electrode was used in this work.

Furthermore, the deviation ratios between CI and CB caused by the fixed TG and CG cells are compared when CB is 5 × 10^−6^ mol/L, as shown in Figure 5D. It was found that there are positive values for both TG and CG cells, but there are still differences between them, which are: Most of the pink columns (the deviation ratios caused by the fixed TG cells) are higher than the gray columns (the deviation ratios caused by the fixed cells in the CG). The explanations for the differences are proposed here. There will be more intracellular I^−^ ions in the TG cells than in the CG cells after the IU procedures, due to the NIS facilitated I^−^ uptake [26]. Subsequently, during the lower C_B_ mediated electronic testing process, the ingested I^−^ may be expelled from the cytoplasm with the help of NIS, which resulted in the increased C_I_ in contrast to C_B_. The analogous differences between TG and CG were also found in the changing ratios of K in Appendix A, and can be interpreted by the similar deduction mentioned above. 

However, there are still exceptions as indicated by the dotted boxes in Figure 5D, for the reason of the small deviation, we speculate that it may have been the interference of Cl^−^, which happens in all kinds of cells [27], which may have lead to the pseudo-responding of I^−^-ISE, as evidenced by our previous work [24]. In addition, the illustrations for the non-zero deviation ratios induced by the fixed CG cells (the gray columns) and NIU cells are also speculated here. One possibility is the osmosis of the intracellular Cl^−^ ions, as mentioned above, the Cl^−^ outflow may cause V_out_ to increase, so the deduced C_I_ values for the CG and NIU cells may be disturbed by the penetration behavior of Cl^−^ ions. Another possibility is the Na^+^ transport, which is known to be simultaneously facilitated by NIS [26], and can still take place even in the absence of I^−^ [28], the discharged Na^+^ ions may accumulate in the PGO scaffold IME and attract I^−^ ions from the background solution to the IME. Accordingly, for the NIU cells immobilized ISE, the deduced C_I_ may be slightly higher than C_B_, as presented by the data of IU-length = 0 in Figure 5D.

## 4. Conclusions

In this work, the function of PGO scaffold IME on ISE for observing the cellular ionic flux is evidenced. The PGO provided spongy-like IME and caused the cellular ion flux to accumulate in this region. Accordingly, there was varied C_I_ in IME, in contrast to the background solution, and the minor ionic fluctuations were aggregated and observed by ISE’s V_out_. Moreover, the correlations of the measured changing ratios of C_I_ in IME due to the NIS related ionic across-membrane transportation were unveiled. In general, it was demonstrated that the PGO decorated ISE, as an ideal combination of the classical ISE’s principle and the modern nanomaterial PGO, can be an alternative for observing across-cytomembrane ion transport. In comparison with the commonly used micro-electrode type devices, the merit of the proposed strategy lies in the independence on the high-cost instruments for fabricating the micro-chamber and measuring the weak transmembrane ionic current.

## Figures and Tables

**Figure 1 sensors-20-03500-f001:**
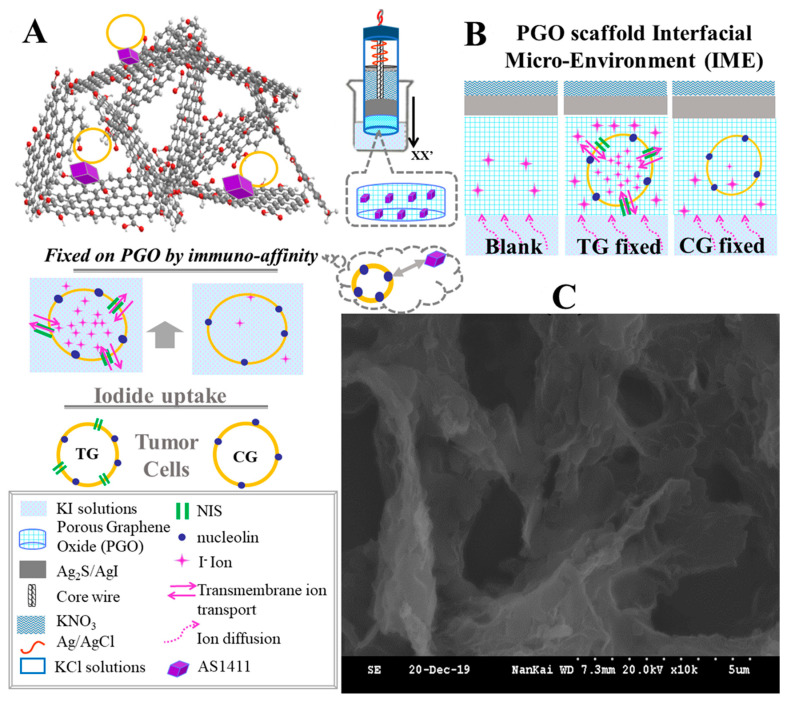
The protocol for observing the across-cytomembrane ion transport induced variations by using PGO decorated I^−^-ISE. (**A**) Schematic diagram of the experimental procedure, that is: The PGO suspension is drop-coated on the sensing membrane (Ag_2_S/AgI) of ISE, then PGO-ISE is functionalized by AS1411 (AS1411-PGO-ISEs); then the tumor cells are incubated on the AS1411-PGO-ISEs by the immune-affinity between AS1411 and nucleolin surface sites on cells’ membrane. These cells are classified in the test group (TG) and the control group (CG), according to their cytomembranes with and without the NIS expression, respectively. The output voltages (V_out_) of ISEs are recorded by immersing them in the similar KI/KNO_3_ buffer solutions. (**B**) The sketch maps of the locally amplified interfacial micro-environment (IME) along the xx’ direction, before and after the cells immobilizations on the AS1411-PGO-ISEs, which are named as Blank PGO, TG fixed and CG fixed, respectively. (**C**) The scanning electron microscope (SEM) of the PGO layer.

**Figure 2 sensors-20-03500-f002:**
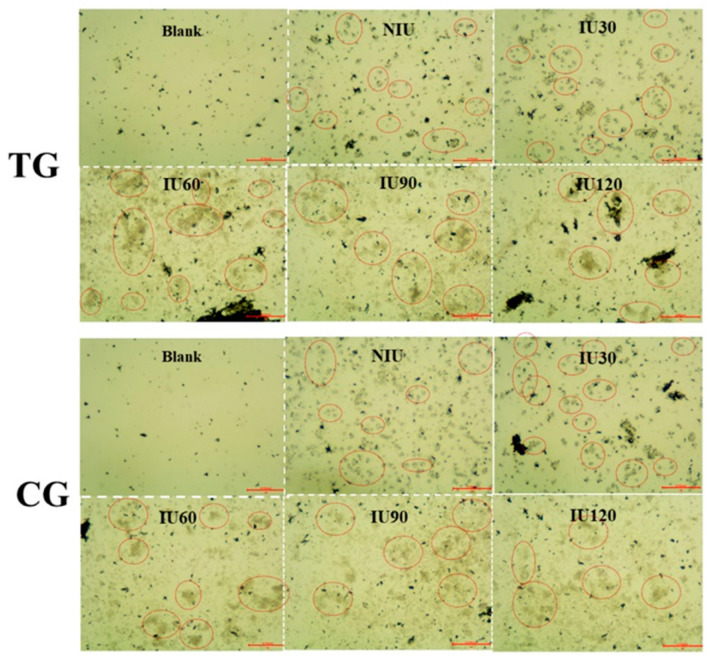
Micrographs of captured cells (MDAMB231 as an example) on the surface of AS1411 and PGO functionalized slides. The photos are classified in two groups, corresponding to the test group (TG) and the control group (CG). Each of them contains two rows, in which the top-right photos are the slides without cells, being used as references to identify the anchored cells in the last five images; meanwhile, the other 5 photos are distinguished from the fixed cells which are without the iodide-uptake treatment (NIU), and treated by IU with various durations 30, 60, 90 and 120 min, labeled as IU30, IU60, IU90 and IU120, respectively.

**Figure 3 sensors-20-03500-f003:**
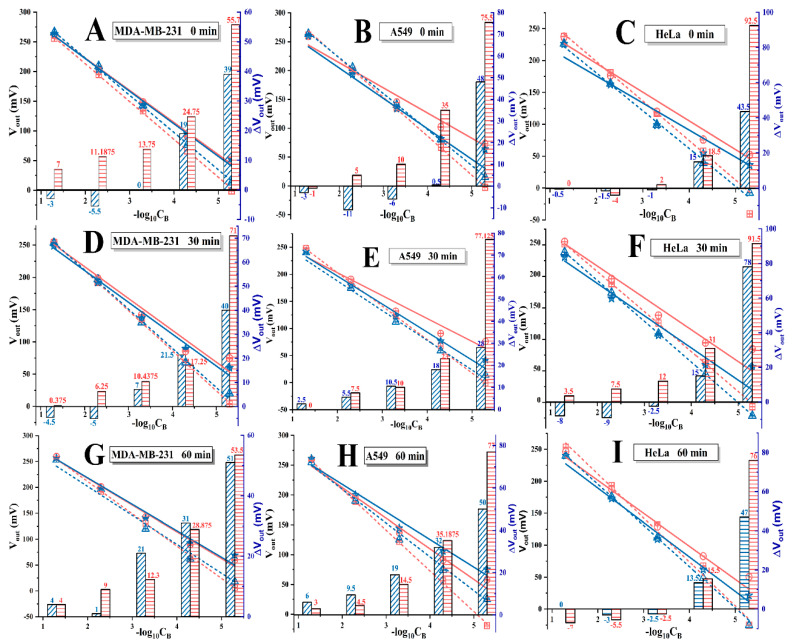
The ISEs responding curves for varied I^−^ concentrations in the background solutions (KI/KNO_3_), before and after they are immobilized by cells, in the tested group (TG) and control group (CG), which are treated by the iodide uptake (IU) operation with the durations of 0, 30, 60 min. The results are arranged in a matrix according to the cells’ type (MDAMB231, A549 or HeLa), and the duration of IU treatment. That is, the first row shows the results of ISEs on immobilized MDAMB231 (**A**), A549 (**B**) and HeLa (**C**) cells without IU treatment (NIU), respectively, the last two rows are the results of the similar ISEs with the only difference in the varied durations of IU treatment ((**D**–**F**) for 30 min, (**G**–**I**) for 60 min). In each of the figures, the histogram shows the changes of the voltage value (ΔV_out_).

**Figure 4 sensors-20-03500-f004:**
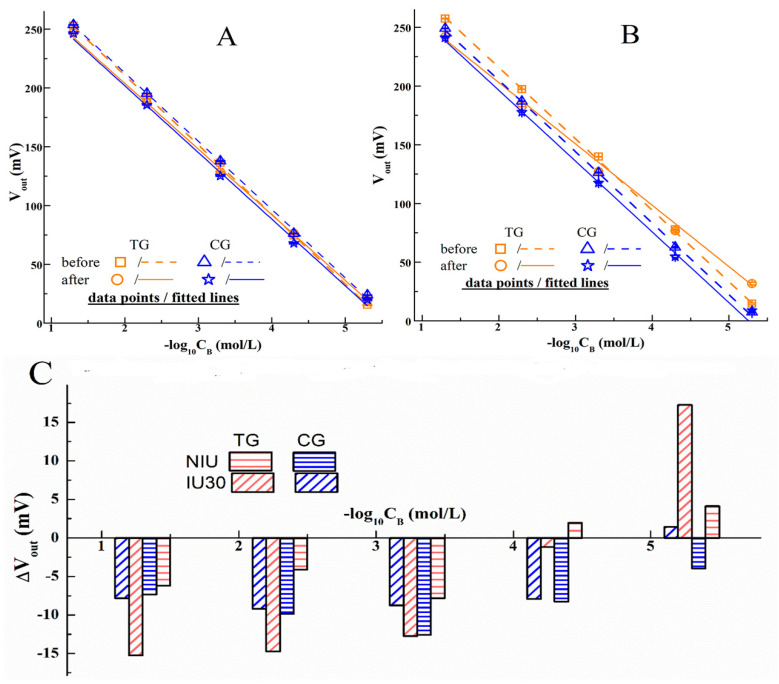
The responding data and the fitted lines of AS1411 functionalized I^−^-ISEs (without PGO) for the KI/KNO_3_ solution for the varied concentrations (C_B_), before and after they are immobilized by cells (MDAMB231) in the test group (TG, the orange symbols and lines) and the control group (CG, the blue ones). (**A**) is the result of ISEs before and after being fixed by the cells without the treatment of IU; (**B**) is the result of ISEs before and after being fixed by the cells with IU operation for 30 min, abbreviated as IU30; (**C**) is the data of the varied V_out_ (ΔV_out_) due to the immobilized cells.

**Figure 5 sensors-20-03500-f005:**
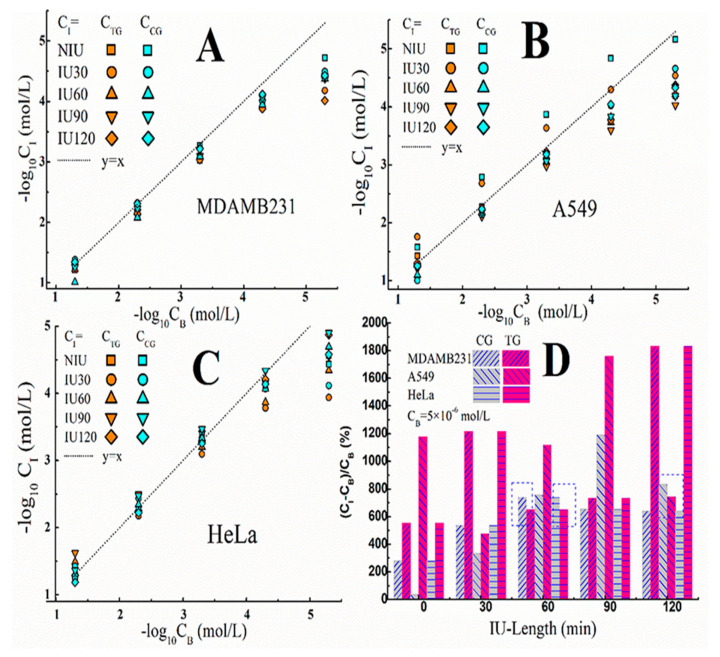
The comparisons of the deduced I^−^ concentrations in IME (C_I_) from Figure 3 in relation with the applied background I^−^ concentration (C_B_). (**A**–**C**) The data for the cells of MDAMB231, A549 and HeLa, respectively. (**D**) The deviation ratios between C_I_ and C_B_ caused by TG and CG cells.

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
