# Peer review of "Porous Graphene Oxide Decorated Ion Selective Electrode for Observing Across-Cytomembrane Ion Transport"

_sensors, 2020, doi:10.3390/s20123500_

Round 1

Reviewer 1 Report

In this paper, it is proposed whether the porous graphene oxide (PGO) film can be used to construct the interface microenvironment (IME) with microporous structure on the surface of traditional ise, so as to detect the transport of iodine ions across the membrane fixed on a large number of cells on the surface of pgo-ise. Taking NIS transfected cells and non NIS transfected cells as examples, the authors carried out a control experiment, and found that NIS transfected cells can cause greater changes in ISE output voltage. In order to verify this hypothesis, the standard curve method was used to calculate the concentration of iodine ion in ime, which was combined with the background buffer concentration used in the experiment. This topic is interesting. The main results are presented clearly. I suggest the manuscript can be published.

Author Response

Response:

We greatly appreciate the comments given by Reviewer #1.

Reviewer 2 Report

The authors experimentally study possibility of usage of the porous graphene oxide (PGO) decorated ion selective electrode (ISE) to observe I- ion transport across cytomembrane of tumor cells. In order to function as a detector, the ISE sensitivity should be sufficiently high. Use of the PGO helps to distribute cells in the pores of PGO (from few to approximately 30 μm) by anchoring with adaptomer AS1411 and optimize the concentration of the measured cells for maximum uptake of the I- ions. The conduction and results of the experiment are described in detail. The increased uptake of I- by the test cells with NIS is clearly demonstrated by ΔVout values in Fig.3 as compared to the control cells. Therefore, the proposed sensor structure can function with sufficient resolution. However, the relative density of anchored with adaptomer cells on the surface of the sensor membrane is estimated only on the micrograph at a single concentration. The cell concentration is estimated in general by CB in solution. Whether the concentration would remain the same as in solution in the PGO and on the sensor surface is not clear from the text. At low concentrations, the linear dependence of the uptake log10CI to log10CB in Fig.5 serves as a confirmation of homogeneous cell density in background solution and sensor surface that lead to the dominance of the transmembrane transport. With the increase in cell concentration, the ion uptake reduces. Authors explain such kind of behavior as the dominance of the ion diffusion mechanism over the trans-membrane ion transport. But the high density of the cells at the sensor surface can lead not only to suppression of transmembrane transport but to some kind of reuptake of I- ions between nearest cells. Therefore, the function of the PGO scaffold IME on ISE help to ensure the optimum cell density at the surface for transmembrane transport at a number of concentrations studied. A discussion of the cell density in the PGO and on ISE vs. background solution could benefit to the general understanding of the conditions of sensing. In Fig.5, the NIU and IU30 marks can be added to the A and B figures directly. The authors clearly demonstrated the possibility to differentiate the tumor cells by measurement of the transmembrane I- transport using PGO decorated ISE.

Recommend for publication with minor adjustments.

Author Response

Response: Our responses are as follows:

  1. As to the cells concentration in the region of PGO mediated IME, the reviewer #2 said it may be interfered by the concentration of I- in the background solution (CB).

   Our opinion is it will not be changed. The reasons are: (1) the cells are anchored on the PGO layer by the immuno-affinity by the similar operation which is described in the manuscript and mentioned at here: 50 μl cell suspension is added on the surface of AS1411-PGO-ISE, after being incubated for 30 min at 37 °C. (2) The variations of CB, which are made in the testing stage, cannot break the affinity theoretically, so it cannot make the fixed cells fall off theoretically and no obvious falling cells are found.

   Even though, it should be admitted, it is very complicated in the real experiments, we really cannot guarantee the number of cells is exactly controlled because of the difficulty in measuring it. We wish this explanation can clarify this comment, and we will be really grateful for your understanding.

  1. As to the high density of cells on the sensor surface, we agree with the comment of Reviewer #2. It may be one of the issues that affect the ionic transmembrane transport. For this reason, we have tried to maintain the amount of the cells on PGO-ISE by controlling the experimental conditions in the incubation process, as mentioned above.
  2. For the function of PGO, we also agree with the reviewer’s idea. So, we add this point in the revised manuscript, at the end of the first paragraph of section 3.3.

    4. For Figure 5, the data about NIU and IU30 are shown in Figure 5A and B, as represented by squares and circles.

Reviewer 3 Report

The authors reported the porous graphene oxide (PGO) anchored on ion selective electrode (ISE) as a detector to detect the signal of cytomembrane ion transport. The changed output voltages of ISEs before and after the cells’ immobilization are in close relation with the sodium-iodide-symporter (NIS) related ion’s across-membrane transportation. And the NIS related minor ionic fluctuations in this spongy-like micro-space can be accumulated and amplified for ISE to probe. Thus, this paper can be accepted after minor revision.

  1. Porous graphene oxide (PGO) was prepared from the hydrothermal method at 180 o In this case, the insulating graphene oxide must be reduced into the conductive reduced graphene oxide, normally called as rGO. Because of the assembly and crosslink, 3D porous rGO was formed in the condition of hydrothermal method. Thus the name of porous graphene oxide is not reasonable.
  2. Generally, graphene oxide from the modified Hummers method is insulated and it can be directly used to register cytomembrane ion transport? The authors should give this comparison.

Author Response

Response:

  1. We agree with this comment, it may be reduced in some degree. But we don’t think it can be classified as rGO, because in our previous work (cited as the reference [24] in the manuscript) we have examined the components in it by XPS. It is demonstrated, though there is an opportunity to be reduced, there are still lots of oxygen-containing functional groups in the PGO prepared by us. So, it is reasonable to believe it in the oxide state.

Yes, graphene oxide from the modified Hummers method is insulated and it can be used to functionalize the sensor surface. While, it may be used for fixing cells, but we still hesitate whether or not it can be directly used to register cytomembrane ion transport. Since, the purpose of this work is to demonstrate the possibility of PGO decorated ISE for observing the cytomembrane ion transport. The idea in this comment may be studied as another special topic in our future.
